# Chinese yuan interest rate swap yields

Tanweer Akram[1]*, Khawaja Mamun[2]

1 Citibank, Irving, Texas, United States of America, 2 Sacred Heart University, Jack Welch College of Business & Technology, Fairfield, Connecticut, United States of America

* tanweer.akram@gmail.com

## Abstract

This paper models the dynamics of Chinese yuan–denominated long-term interest rate swap yields. It shows that the short-term interest rate exerts a decisive influence on the long-term swap yield after controlling for various macrofinancial variables, such as core inflation, the growth of industrial production, the percent change in the equity price index, and the percentage change in the Chinese yuan exchange rate. The autoregressive distributed lag approach is applied to model the dynamics of the long-term swap yield. The findings reinforce and extend John Maynard Keynes's conjecture that in advanced countries, as well as emerging market economies such as China, the central bank's actions have a decisive role in setting the long-term interest rate on government bonds and over-the-counter financial instruments, such as swaps.

**Data Availability Statement:** All relevant data are within the paper and its Supporting Information files.

**Funding:** The authors received no specific funding for this work.

## Introduction

This paper econometrically models the dynamics of Chinese yuan (CNY)–denominated long-term interest rate swap yields using monthly macroeconomic and financial data. The financial sector plays a vital role in the Chinese economy, which has grown rapidly in the past several decades. There has also been a rapid growth of outstanding debt and fixed-income instruments, with notable developments in interest rate liberalization, accompanied by a spectacular rise in the country's bond market and total social financing since the global financial crisis.

China's gross domestic product (GDP) per capita, measured in terms of the 2015 US dollar, has risen from merely $430 in 1980 to $11,200 in 2021 (Fig 1), according to [1]. Between 1980 to 2022, the country's GDP growth averaged slightly over 9 percent per year. China is now an upper-middle-income country. Thanks to the country's rapid growth, more than 800 million people have been lifted out of poverty. China has successfully eradicated extreme poverty, as measured by the number of people living on less than $1.90 per day. There have also been significant improvements in access to health, education, and various other indicators of development, according to [2]. The country has invested substantially and rapidly improved its infrastructure. Meanwhile, China's financial sector has grown impressively along with its economy's rapid growth and transformation. Domestic credit to the private sector, as a share of GDP, has also risen markedly from merely 53 percent of GDP in 1980 to slightly more than 180 by 2020 (Fig 2), based on [1]. This denotes a marked rise in the financialization of the Chinese economy.

**Competing interests:** The authors have declared that no competing interests exist.

Interest rate swaps are likely to play an important role in the Chinese financial system, which has been changing from a bank-dominated system to one with more diverse financial institutions and increased market dominance. Although there is a growing literature studying the Chinese financial system [3, 4], CNY-denominated interest rate swap yields have not been econometrically modeled in the existing literature. The analysis of CNY-denominated swaps warrants careful study because of the increased financialization of the Chinese economy [3, 4] and the rise of the nation's shadow banking system in which over-the-counter derivatives, such as swaps, are likely to have a crucial role.

This paper econometrically models the evolution of CNY swap yields (Fig 3). It shows that the short-term interest rate exerts a decisive influence on the long-term swap yield after controlling for various macroeconomic and financial variables using an autoregressive distributed lag (ARDL) model. The effect of short-term interest rate on the CNY swap yield declines as the maturity tenor of the swap increases. The study also finds that the 6-month interest rate has a larger effect on the swap yield than the 3-month interest rate. These findings are in concordance with John Maynard Keynes's [5, 6] astute insight into the relationship between the long-term interest rate and the current short-term interest rate. The findings reinforce and extend Keynes's notion that the central bank's actions play a decisive role in setting the long-term interest rate, not just in advanced countries but also in emerging markets, and not just for government bonds but also for over-the-counter financial instruments.

The paper proceeds as follows. Section II provides a short primer on interest rate swaps and briefly reviews the relevant literature on swaps. Section III presents a simple model of the swap yield that relates the dynamics of the swap yield to the short-term interest rate and some key macroeconomic and financial variables. Section IV discusses the data and sources used in the econometric modeling of swaps yields and undertakes unit root and stationarity tests. Section V lays out the framework for the econometric models, reports and interprets the findings from the estimated models, and discusses the implications of the results. Section VI concludes.

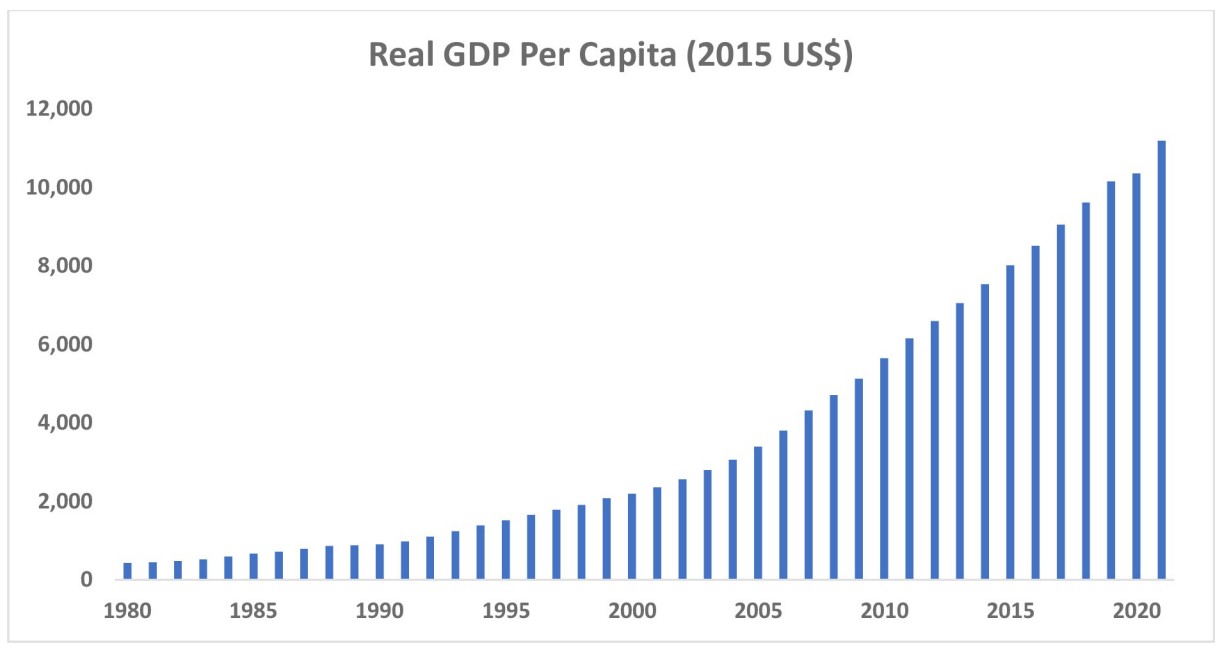

**Fig 1. Per capita GDP (measured in 2015 USD) has risen sharply in China, 1980–2021.** Source: [1].

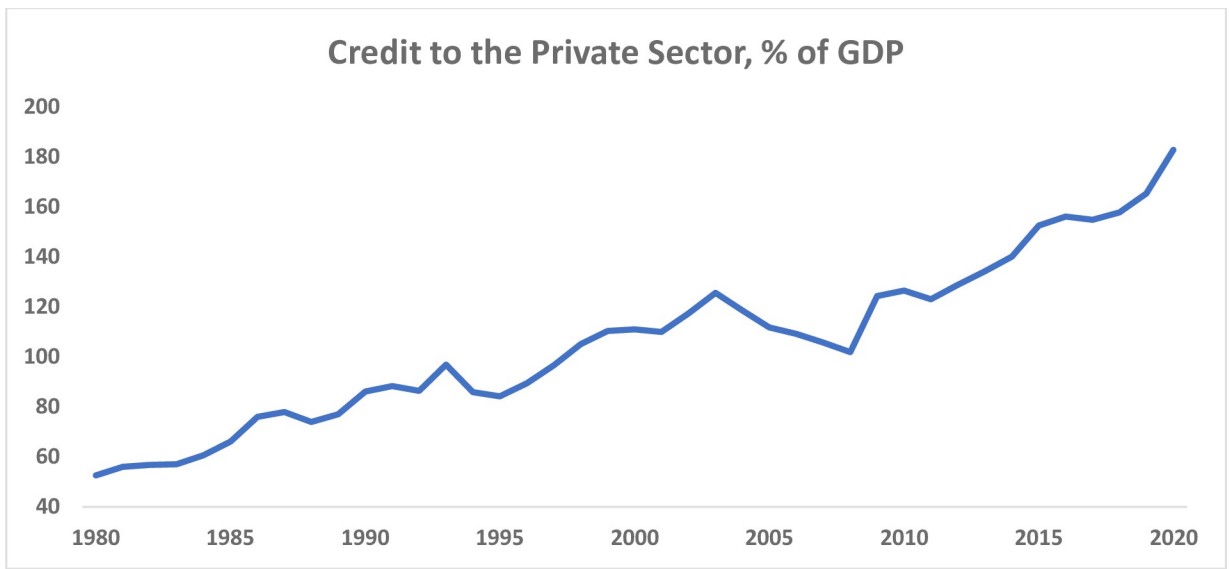

**Fig 2. The evolution of domestic credit to the private sector, 1980–2020.** Source: [1].

## Interest rate swaps and a brief review of the literature

Interest rate swaps are contracts that enable two parties to exchange two interest rate cash flows with different features. Swaps are derivative contracts that trade over the counter. The principal amount is the same for both parties. For plain-vanilla interest rate swaps, the swap buyer pays the fixed interest rate and receives the variable interest rate. The buyer is known as the receiver. The swap seller pays the variable interest rate and receives the fixed interest rate. The swap seller is known as the payer. The floating rate payments are based on some benchmark interest rates plus some agreed-upon markup. The swap yield or the swap rate is the

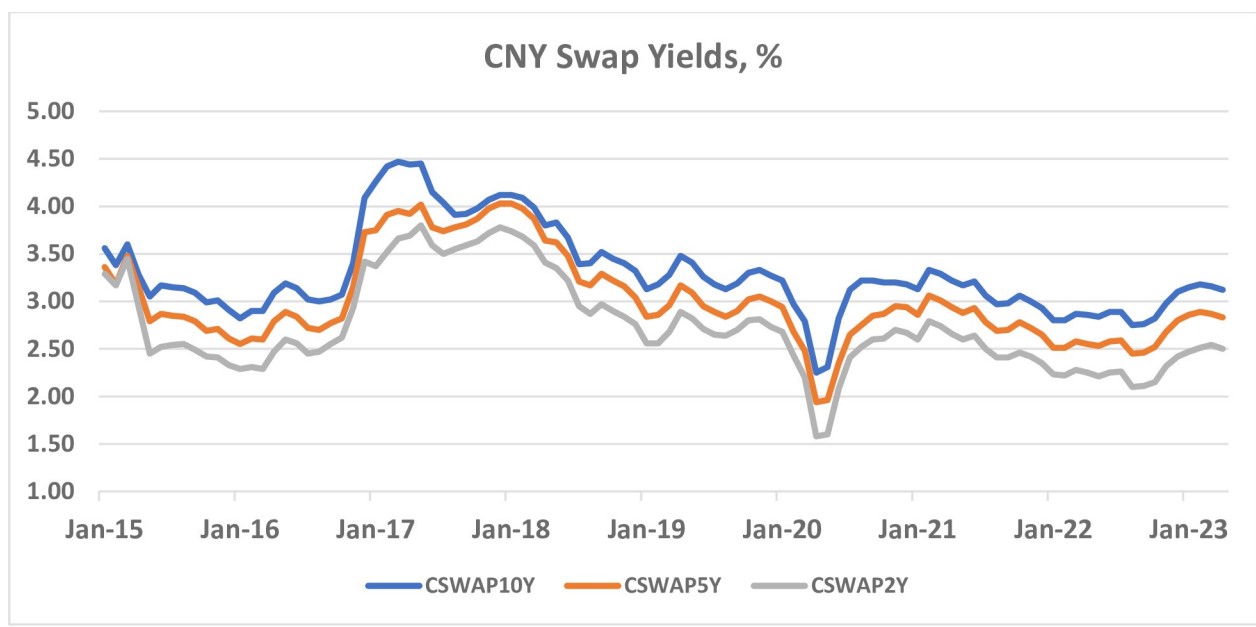

**Fig 3. The evolution of CNY swap yields, January 2015–April 2022.** Source: Listed in Table 1.

fixed interest rate that the buyer or the receiver demands in exchange for the uncertainty of having to pay for the variable interest rate based on some short-term benchmark interest rate, which changes over time. Swaps are usually quoted in terms of the fixed interest rate.

[7] explains the functions of interest rate swaps, including usage, pricing, risks, and innovations. [8–12] give overviews of the assorted use of swaps in various business and finance applications. Though there is a vast literature on swaps, the empirical modeling of swap yields has some critical gaps. [13–15] have pioneered the empirical modeling of swap yields, but these models fail to incorporate Keynes's [5, 6] insight, which tethers the long-term interest rate to the short-term interest rate. [13] relates swap yields to credit quality, while [14] unravels swap yields in terms of credit and liquidity factors. [15] examines the swap spread with respect to government bond yields, rather than swap yields; the authors report finding procyclical behavior for the short maturity U.S. swap spreads and countercyclical behavior for longer maturity U.S. swap spreads, while for the United Kingdom, swap spreads are countercyclical across maturities. [16] explains swap yields or swap spreads with reference to the aggregated funding status of defined benefit plans and relates this to the swap spreads at the back end of the swap yield curve. The predominant approach in modeling swap yields has been through microeconomic and financial analysis, rather than a macroeconomic approach. The previous empirical literature on swap yield dynamics has been most constricted to the financial markets of advanced countries, such as the United States and the United Kingdom.

While previous empirical literature on swap yields and Treasury-to-swap spreads has attempted to explain the dynamics in terms of liquidity and credit quality [13, 14] and the underfunding of pension benefits plans (such as [16]), the econometric analysis undertaken here shows the short-term interest rate is the main driver of the swap yields of various tenors, after controlling for key macroeconomic and financial variables, such as core inflation or total inflation, the growth of industrial production, the percentage change in the equity price, and the percentage change in the exchange rate. The findings vindicate Keynes's view of interest rate dynamics and extend it to long-term swap yields, an over-the-counter financial instrument. Nevertheless, it must be acknowledged that liquidity, credit conditions, and balance sheet positions can and do influence swap yields.

[17] shows Keynes's insight drew on his own theoretical perspectives and Riefler's [18] pioneering statistical analysis of bond yields in the United States in the 1920s. Keynes's insight on interest rate dynamics has also found support in recent empirical research. Several studies, such as [19–31], evince that there is a meaningful economic and statistically significant pass-through from the central bank's policy rate to market interest rates. [32, 33] have advanced quantitative models that formalize Keynes's insight linking the long-term interest rate to the short-term interest rate.

Most empirical studies showing the strong connection between the short-term and long-term interest rates have been confined to government bond yields in advanced countries. Hence, it is relevant to ask whether the Keynesian insight is generalizable beyond government bond yields in advanced countries. This paper contributes to the empirical literature by examining: (i) whether Keynes's conjecture is applicable to spread products and over-the-counter financial derivatives, such as interest rate swaps, and (ii) whether it holds in emerging markets with a rapidly evolving financial system, such as China.

Recently [34–38] there has been a nascent flourishing of empirical studies on swap yields from a Keynesian perspective using assorted econometric methods. These studies [34–38] have shown that Keynes's conjecture is supported for swaps denominated in both hard currencies, such as the British pound sterling (GBP), US dollar (USD), and Japanese yen (JPY), and emerging market currencies, such as the Chilean peso (CLP) and Indian rupee (INR). [34] relies on ARDL models to report that for USD swaps, the monthly changes in the Treasury bill

rate exert a statistically significant effect on the monthly changes in swap yields of different maturity tenors after controlling for a host of macroeconomic and financial variables. [35] applies generalized autoregressive conditional heteroskedasticity (GARCH) models for GBP swaps and reports that the month-over-month change in the short-term interest rate has a positive and statistically significant effect on the month-over-month change in long-term swap yields, after controlling for changes in inflationary pressure, the change in the growth of industrial production, the percentage change in the equity price index, and the percentage change in the GBP exchange rate. [36] examines JPY-denominated interest rate swaps, finding evidence of structural breaks in the dynamics of swap yields using Bai-Peron tests. The authors report that the short-term interest rate exerts an important influence on the long-term swap yield in some periods but not in other periods in which core inflation exerts a marked influence on the swap yield. The findings from the econometric models reveal that a discernable relationship between the call rate and the swap yield of different maturity tenors clearly held prior to April 2014 but did not in the subsequent period.

The Keynesian approach to modeling swap yields has also been recently extended to selected emerging markets. [37] implements a GARCH approach to modeling the dynamics of the long-term swap yield for CLP swaps. It reports that the change in the short-term interest rate has an economically meaningful and statistically significant effect on the change in the interbank swap yield, after controlling for key macroeconomic and financial variables, including inflation, the growth of industrial production, and the percentage change in both the equity price index and the exchange rate. The authors assert that their findings imply that the Banco Central de Chile's monetary policy exerts an important influence on interbank swap yields in Chile. [38] ventures to apply ARDL models to INR swaps. The authors state that they find that the short-term interest rate has an important influence on swap yields, after controlling for a standard list of macroeconomic and financial variables, including inflation, industrial production, the percentage change in the equity price index, and the percentage change in the INR exchange rate. The authors claim that their findings imply that the Reserve Bank of India can sway borrowing and lending rates not just on Indian government bonds but for also INR-denominated private-market financial instruments. There is no comparable study of the dynamics of CNY swaps, even though their usage has grown in CNY-denominated financial markets in China and overseas. Research of the major databases, including JSTOR, ECONLIT, Google Scholar, PROQUEST, SSRN, and so forth, shows that there are no econometric studies of CNY-denominated interest rate swap yields. Thus, this paper fills a critical lacunae in the literature.

## A simple model of swap yields

A simple model of the swap yield is presented here. The model presented here revises and attenuates [32, 33]'s interest rate models pertaining to government bond yields to make them suitable for analyzing the dynamics of the long-term swap yield. The long-term interest rate swap yield is $S_{LT}(t)$. The short-term interest rate is $i_{ST}(t)$. The central bank's policy rate is $i_{CB}$. Inflation is $\pi(t)$, while the central bank's inflation target is $\bar{\pi}$. $\chi(t)$ represents financial market volatility, while $\tau(t)$ is an exogenous shock. $W_1(t)$, $W_2(t)$, $W_3(t)$, and $W_4(t)$, are distinct Weiner processes. The parameters of the models are: $\alpha_1$, $\alpha_2$, $\beta$, $\gamma$, and $\delta$.

$$dS_{LT}(t) = (\alpha_1 i_{ST}(t) + \alpha_2 \pi(t))dt + \chi(t)\sqrt{i_{ST}(t)}dW_1(t) \tag{1}$$

$$di_{ST}(t) = \beta(i_{CB}(t) - i_{ST}(t))dt + \chi(t)\sqrt{i_{ST}(t)}dW_2(t) \tag{2}$$

$$d\pi(t) = \gamma(\bar{\pi} - \pi(t))dt + \chi(t)\sqrt{\pi(t)}dW_3(t) \tag{3}$$

$$d\chi(t) = \delta(\bar{\chi} - \chi(t))dt + \tau(t)\sqrt{\chi(t)}dW_4(t) \tag{4}$$

Eq (1) relates the dynamics of the long-term swap yield to: (i) the change in the sum of the short-term interest rate and inflation, and (ii) the change in a Weiner process adjusted by the volatility of the financial market and the short-term interest rate. Eq (2) expresses the dynamics of the short-term interest rate as a function of: (i) the difference between the central bank's policy rate and the short-term interest rate, and (ii) a Weiner process adjusted by the volatility of the financial market and the short-term interest rate. Eq (3) relates the dynamics of inflation to: (i) the difference between the central bank's inflation target and the observed inflation, and (ii) a Weiner process adjusted by the volatility of the financial market and inflation. Eq (4) connects the dynamics of the volatility of the financial market to: (i) a mean reverting process, and (ii) a Weiner process adjusted by an exogenous shock and the volatility of the financial market.

The above model ties the dynamics of the interbank swap yield to fundamental macroeconomic and financial variables, such as the change in the short-term interest rate, the change in inflation, and the volatility of the financial market and some Weiner processes. This model can be extended to incorporate other relevant factors, such as the change in the growth of industrial production, the change in the logarithm of the equity price index, and the change in the logarithm of the exchange rate, if these factors are deemed as important drivers of the interbank swap yield.

## Data, summary statistics, and unit root and stationarity tests

Table 1, below, describes the data used in the paper. Swap yields of three different maturity tenors are used. (Swap yields are measured in the standard manner.

The swap yield denotes the fixed rate that a party to a swap contract requests in exchange for the obligation to pay a variable interest rate, typically based on a benchmark interest rate agreed upon by the contracting parties). Two measures of the short-term interest rate are used, that is, Treasury bill rates of 3-month and 6-month tenors. Inflation is measured as the year-over-year percent change in: (i) the consumer price index (CPI) excluding food and energy, which is regarded as core inflation, and (ii) the CPI, which is regarding as headline or total inflation. Economic activity is calibrated from the year-over-year growth of industrial production. Equity price is based on the index of the: (i) Shanghai Stock Exchange (SSE) and (ii) Shenzhen Stock Exchange (SZSE). Two different measures of the exchange rate are: (i) the value of the CNY per USD and (ii) the value of CNY per euro. In the text and tables below, LN (.) indicates the (natural) logarithm of a variable. The monthly time-series data cover from September 2014 to April 2023. Thus, each time series consists of 104 observations. The time period that has been selected for empirical modeling is based on: (i) the introduction and the widespread adoption and use of CNY-denominated interest rate swaps for both hedging and speculating on interest rate risks by domestic and overseas parties, and (ii) the availability of the data. The data used in the paper are available in the attached file: S1 Dataset.

The summary statistics of all variables in their levels and first differences are presented in Tables 2 and 3, respectively. The average swap yield increases with the maturity tenors of the swap, as higher maturity indicates higher risk. Similarly, the mean of the 6-month Treasury bill rate is higher than the mean of the 3-month Treasury bill rate. The Jarque-Bera tests indicate that higher maturity swap yields, inflation and core inflation, and the growth of industrial production are not normally distributed.

Table 3 shows the summary statistics of all the variables at their first difference. The short-run interest rates and swap yields are more volatile at their first difference. None of the

**Table 1. Data description.**

| Variable label | Description, date range | Frequency | Sources |
|---|---|---|---|
| *Swap yields* | | | |
| SWAP2Y | Interest rate swap, 2 years, CNY, % September 2014–April 2023 | Daily; converted to monthly | Tullet Prebon Information |
| SWAP5Y | Interest rate swap, 5 years, CNY, % September 2014–April 2023 | Daily; converted to monthly | Tullet Prebon Information |
| SWAP10Y | Interest rate swap, 10 years, CNY, %, September 2014–April 2023 | Daily; converted to monthly | Tullet Prebon Information |
| *Short-term interest rates* | | | |
| CTB3M | Treasury bill, 3 months, %, September 2014– April 2023 | Daily; converted to monthly | People's Bank of China |
| CTB6M | Treasury bill, 6 months, %, September 2014–April 2023 | Daily; converted to monthly | People's Bank of China |
| *Inflation* | | | |
| CCPIYOY | Consumer price index excluding food and energy, %, change, y/y, September 2014–April 2023 | Monthly | China National Bureau of Statistics |
| CPIYOY | Consumer price index, % change, y/y, September 2014–April 2023 | Monthly | China National Bureau of Statistics |
| *Economic activity* | | | |
| IPYOY | Industrial production: % change, y/y, September 2014–April 2023 | Monthly | China National Bureau of Statistics |
| *Financial market* | | | |
| SNGHAI | Shanghai Stock Exchange, stock price index, close price, September 2014–April 2023 | Daily; converted to monthly | Shanghai Stock Exchange |
| SNZN300 | Shanghai-Shenzhen 300, Stock price index, close price, September 2014–September 2022 | Daily; converted to monthly | Shanghai Stock Exchange |
| *Exchange rate* | | | |
| USDCNY | Exchange rate, ¥/US$, average, September 2014–April 2023 | Daily; converted to monthly | Federal Reserve Board |
| EURCNY | Exchange rate, ¥/€, average, September 2014–April 2023 | Daily; converted to monthly | European Central Bank |

**Table 2. Summary statistics of the variables.**

| Vars | Obs | Mean | Std. Dev. | Max | Min | Skewness | Kurtosis | J-B | Probability |
|---|---|---|---|---|---|---|---|---|---|
| SWAP2Y | 104 | 2.75 | 0.50 | 3.80 | 1.58 | 0.51 | 2.64 | 4.98 | 0.08 |
| SWAP5Y | 104 | 3.03 | 0.47 | 4.03 | 1.94 | 0.61 | 2.77 | 6.75 | 0.03 |
| SWAP10Y | 104 | 3.31 | 0.46 | 4.47 | 2.25 | 0.80 | 3.32 | 11.49 | 0.00 |
| CTB3M | 104 | 2.34 | 0.56 | 3.97 | 1.00 | 0.57 | 3.14 | 5.77 | 0.06 |
| CTB6M | 104 | 2.46 | 0.55 | 3.92 | 1.10 | 0.51 | 3.02 | 4.57 | 0.10 |
| CPIYOY | 104 | 1.87 | 1.00 | 5.85 | -1.13 | 0.79 | 6.17 | 54.45 | 0.00 |
| CCPIYOY | 104 | 1.36 | 0.58 | 2.31 | 0.10 | -0.38 | 2.08 | 6.12 | 0.05 |
| IPYOY | 104 | 5.96 | 5.14 | 33.94 | -13.91 | 2.03 | 21.43 | 1542.69 | 0.00 |
| LNSNGHAI | 104 | 7.88 | 0.15 | 8.29 | 7.59 | 0.26 | 2.47 | 2.34 | 0.31 |
| LNSNZN300 | 104 | 8.26 | 0.17 | 8.62 | 7.79 | -0.08 | 2.95 | 0.13 | 0.94 |
| LNUSDCNY | 104 | 1.89 | 0.04 | 1.97 | 1.81 | -0.10 | 1.94 | 5.06 | 0.08 |
| LNEURCNY | 104 | 2.02 | 0.05 | 2.10 | 1.90 | -0.53 | 2.32 | 6.92 | 0.03 |

**Table 3. Summary statistics of the first differences of the variables.**

| Vars | Obs | Mean | Std. Dev. | Max | Min | Skewness | Kurtosis | J-B | Probability |
|---|---|---|---|---|---|---|---|---|---|
| ΔSWAP2Y | 103 | -0.01 | 0.16 | 0.49 | -0.62 | -0.38 | 6.18 | 45.92 | 0.00 |
| ΔSWAP5Y | 103 | -0.01 | 0.16 | 0.58 | -0.54 | 0.15 | 5.52 | 27.65 | 0.00 |
| ΔSWAP10Y | 103 | -0.01 | 0.16 | 0.69 | -0.54 | 0.65 | 7.08 | 78.76 | 0.00 |
| ΔCTB3M | 103 | -0.02 | 0.24 | 0.70 | -0.75 | -0.30 | 4.01 | 5.89 | 0.05 |
| ΔCTB6M | 103 | -0.02 | 0.21 | 0.75 | -0.61 | 0.01 | 4.95 | 16.35 | 0.00 |
| ΔCPIYOY | 103 | -0.01 | 0.51 | 1.39 | -1.62 | -0.43 | 4.23 | 9.79 | 0.01 |
| ΔCCPIYOY | 103 | -0.01 | 0.13 | 0.40 | -0.40 | 0.19 | 4.08 | 5.57 | 0.06 |
| ΔIPYOY | 103 | -0.02 | 4.33 | 26.85 | -20.87 | 0.43 | 25.54 | 2183.45 | 0.00 |
| ΔLNSNGHAI | 103 | 0.00 | 0.06 | 0.19 | -0.26 | -0.86 | 7.29 | 91.73 | 0.00 |
| ΔLNSNZN300 | 103 | 0.01 | 0.06 | 0.23 | -0.22 | -0.22 | 6.94 | 67.49 | 0.00 |
| ΔLNUSDCNY | 103 | 0.00 | 0.01 | 0.04 | -0.03 | 0.48 | 4.29 | 11.15 | 0.00 |
| ΔLNEURCNY | 103 | 0.00 | 0.02 | 0.04 | -0.05 | -0.22 | 3.98 | 4.95 | 0.08 |

Note: One observation is lost due to the first-differencing of the data.

variables have a normal distribution, according to the Jarque-Bera tests. The change in the growth of industrial production shows a large decline in March 2021, indicating the impact of the pandemic lockdowns on China's industrial sector.

The unit root and stationarity tests' results are given in Tables 4 and 5. They present both augmented Dickey-Fuller (ADF) unit root tests, as described in [39, 40], and Kwiatkowski-Phillips-Schmidt-Shin (KPSS) stationarity tests, based on [41]. The null hypotheses for the ADF and the KPSS tests are different. The null hypothesis of the ADF test is that the time series contains a unit root, while the alternative hypothesis is that the time series does not contain a unit root. The null hypothesis of the KPSS test is that the time series is stationary, while the alternative hypothesis is that the time series is nonstationary.

**Table 4. Unit root and stationarity tests of the variables.**

| Variables at level | ADF unit root tests ($H_0$: has unit root) | | | KPSS tests ($H_0$: stationarity) | |
|---|---|---|---|---|---|
| | None | Intercept | Trend | Intercept | Trend |
| SWAP2Y | -0.65 | -2.31 | -2.60 | 0.45* | 0.12 |
| SWAP5Y | -0.56 | -2.25 | -2.46 | 0.41* | 0.12* |
| SWAP10Y | -0.48 | -2.56 | -2.83 | 0.40* | 0.12* |
| CTB3M | -1.28 | -3.21** | -3.59** | 0.61** | 0.08 |
| CTB6M | -1.38 | -2.86* | -3.09 | 0.52** | 0.09 |
| CPIYOY | -0.66 | -1.51 | -1.59 | 0.09 | 0.09 |
| CCPIYOY | -0.96 | -0.60 | -1.79 | 0.79*** | 0.19** |
| IPYOY | -2.06** | -3.66*** | -3.65** | 0.06 | 0.05 |
| LNSNGHAI | 0.68 | -2.42 | -2.44 | 0.43* | 0.16** |
| LNSNZN300 | 0.79 | -2.93** | -2.78 | 0.74*** | 0.07 |
| LNUSDCNY | 0.65 | -2.73* | -2.90 | 0.36* | 0.14* |
| LNEURCNY | -0.33 | -2.00 | -2.10 | 0.25 | 0.22*** |

**Note:** Significance levels

*** for 1 percent

** for 5 percent, and

* for 10 percent

**Table 5. Unit root and stationarity tests of the first differences of the variables.**

| Variables at first difference | ADF unit root tests (H₀: has unit root) | | | KPSS tests (H₀: stationarity) | |
|---|---|---|---|---|---|
| | None | Intercept | Trend | Intercept | Trend |
| ΔSWAP2Y | -7.49*** | -7.46*** | -7.41*** | 0.07 | 0.07 |
| ΔSWAP5Y | -7.74*** | -7.70*** | -7.65*** | 0.08 | 0.07 |
| ΔSWAP10Y | -7.25*** | -7.21*** | -7.16*** | 0.06 | 0.06 |
| ΔCTB3M | -8.15*** | -8.14*** | -8.12*** | 0.07 | 0.05 |
| ΔCTB6M | -5.58*** | -5.59*** | -5.57*** | 0.09 | 0.06 |
| ΔCPIYOY | -5.48*** | -5.44*** | -5.45*** | 0.08 | 0.05 |
| ΔCCPIYOY | -9.84*** | -9.84*** | -9.91*** | 0.14 | 0.06 |
| ΔIPYOY | -10.98*** | -10.92*** | -10.86*** | 0.15 | 0.15* |
| ΔLNSNGHAI | -6.42*** | -6.42*** | -6.39*** | 0.07 | 0.07 |
| Δ LNSNZN300 | -8.23*** | -8.24*** | -8.29*** | 0.14 | 0.07 |
| ΔLNUSDCNY | -5.98*** | -6.00*** | -5.98*** | 0.08 | 0.07 |
| ΔLNEURCNY | -8.34*** | -8.30*** | -8.27*** | 0.09 | 0.09 |

**Note:** Significance levels

*** for 1 percent

** for 5 percent, and

* for 10 percent

Table 4 exhibits the unit root and stationarity tests of the variables at the level. The ADF tests show that for most variables the hypothesis that it contains a unit root cannot be rejected. The KPSS tests reveal that for most of the variables the hypothesis it is stationary can be rejected. The exception to this generalization is the growth of industrial production.

Table 5 shows the ADF and KPSS tests of the variables in their first difference. The ADF tests reject the null hypothesis of a unit root for all variables in their first difference. The KPSS tests cannot reject the null hypothesis of stationarity in most cases. In a few cases, KPSS tests reject the null hypothesis of stationarity. Based on ADF and KPSS tests, the overall picture provides pretty strong support for: (i) the rejection of the presence of a unit root and (ii) the failure to reject the null hypothesis of stationarity for most of these variables at the first difference.

## Econometric framework, findings of the estimated models, and interpretations

**Econometric framework.** Given the time-series properties of the data that was examined in the previous section, the ARDL approach is the most appropriate for modeling the dynamics of CNY interest rate swaps, as the variables are either stationary, I(0), or integrated in the first order, I(1). Estimates based on the ARDL approach can reveal both the short-run and long-run effects of the independent variables on the swap yield.

Three different models for each maturity tenor of the swaps are estimated. In the simple model, the swap yield is just a function of the short-term interest rate. In the basic model, the swap yield is a function of the short-term interest rate, core inflation, and the growth of industrial production. In the extended model, the swap yield is a function not just of the short-term interest rate, core inflation, and the growth of industrial production, but also of the month-over-month percentage change in the equity price index and the month-over-month percentage change in the exchange rate. For each model, the swap yields of three different maturity tenors—namely 2-year, 5-year, and 10-year tenors—are used as the dependent variable in the

regression equations. The empirical equation for the extended model can be described as:

$$CSWAPiY_t = \alpha_0 + \beta_1 CTB3M_t + \beta_2 CTB3M_{t-1} + \beta_3 CTB3M_{t-2} + \beta_4 CSWAPiY_{t-1}$$
$$+ \beta_5 CSWAPiY_{t-2} + \beta_6 CCPI_t + \beta_7 IPYOY_t + \beta_8 LNSNGHAI_t + \beta_9 LNUSDCNY_t$$
$$+ \epsilon_t$$

where i = swap yields for swaps for 2-, 5-, and 10-year tenors.

## Econometric results

The main results are displayed in Tables 6–8. In all models with three different maturity levels of swap yields, the yield of the 3-month Treasury bill has a positive and statistically significant effect on the swap yield. A 100–basis point increase in the 3-month Treasury bill rate increases the 2-year swap yield by 43–44 basis points. (A basis point is one one-hundredth of one percent. Hence, 43–44 basis points means 0.43–0.44 percentage points. Basis points are usually used when discussing interest rates and interest rate spreads.) The effect declines with a higher maturity tenor for the swap. The effect of the Treasury bill rate declines to 35–37 basis points for a 10-year swap. The long-run relationship between the 3-month interest rate and the swap yield declines significantly from the 2-year to the 10-year maturity term. In addition, the relationship weakens when more control variables are added for swaps at all three term lengths. The rate of adjustment to any shock to the long-run relationship between the Treasury bill rate and the swap yield differs for different maturity tenors. A shock dissipates somewhere between five months to seven months, after which the relation between the Treasury bill rate and the swap yield returns to it long-run equilibrium.

The lagged values of 3-month Treasury rates affect the CNY swap yield significantly up to two months lagged. In addition, the lagged value to the swap rates also has significant effects up to two months in the past. Thus, the ARDL models show that both the 3-month short-term interest rate and the lagged dependent variables have a somewhat long memory. Among the control variables, the core inflation rate, the growth of industrial production, and the percentage change in the SSE have a positive impact on the swap yield. A higher level of core inflation requires a higher swap yield, whereas stronger growth in industrial production and/or a rise in the equity index leads to a higher swap yield.

The adjusted $R^2$ of these models implies that much of the variance in the swap yield is explained by the Treasury bill rate and its lags, as well as the autoregressive variables. The Akaike Information Criterion (AIC) also shows a good fit for all the models. A panel of postestimation diagnostic tests is also displayed in Tables 6–8. The joint significance tests reveal a strong rejection of the insignificance of the regressors. The Durbin-Watson statistics and Breusch-Godfrey Lagrange Multiplier tests indicate there is no serial correlation in the error terms in these models. The correlogram Q-statistics, which are available upon request, imply that the mean equations in these models are correctly specified and there are no remaining serial correlations. The Breusch-Pagan-Godfrey heteroskedasticity tests fail to reject the null hypothesis of homoscedasticity in all models except one, at the 5 percent significance level. The Jarque-Bera tests indicate that the error terms are not normally distributed, which is a not an uncommon phenomenon for financial variables. Last but not least, the Ramsey RESET tests indicate all the models are well specified. Additional parameter stability tests using CUMSUM and CUMSUM-SQ, as articulated in [42], are available upon request.

**Sensitivity analysis.** Sensitivity analysis is conducted by changing some key independent variables in the regression models. The results of the sensitivity analysis are displayed in Tables 9–11. The empirical equations of the extended model with alternative independent variables

**Table 6. ARDL(p, q) model of SWAP2Y.**

| Dependent variable | SWAP2Y | SWAP2Y | SWAP2Y |
|---|---|---|---|
| **Main equation** | | | |
| **CTB3M** | 0.44*** | 0.43*** | 0.43*** |
| | (0.00) | (0.00) | (0.00) |
| **CTB3M(-1)** | −0.49*** | −0.49*** | −0.48*** |
| | (0.00) | (0.00) | (0.00) |
| **CTB3M(-2)** | 0.19*** | 0.25*** | 0.17** |
| | (0.00) | (0.00) | (0.01) |
| **SWAP2Y(-1)** | 1.10*** | 1.09*** | 1.09*** |
| | (0.00) | (0.00) | (0.00) |
| **SWAP2Y(-2)** | −0.28** | −0.30** | −0.29* |
| | (0.05) | (0.04) | (0.05) |
| **Core CPIYOY** | | 0.05* | 0.05** |
| | | (0.10) | (0.04) |
| **IPYOY** | | 0.004** | 0.004** |
| | | (0.01) | (0.01) |
| **ΔLNSNGHAI** | | | 0.34 |
| | | | (0.22) |
| **ΔLNUSDCNY** | | | −0.78 |
| | | | (0.37) |
| **Intercept** | 0.16** | 0.20** | 0.20*** |
| | (0.02) | (0.01) | (0.00) |
| **Cointegrating relationship** | | | |
| **Long-term coefficient** | 0.79*** | 0.58*** | 0.54*** |
| | (0.00) | (0.00) | (0.00) |
| **Rate of adjustment** | −0.18*** | −0.21*** | −0.20*** |
| | (0.00) | (0.00) | (0.00) |
| **Model information** | | | |
| **Obs** | 102 | 102 | 102 |
| **Adj R$^2$** | 0.94 | 0.94 | 0.94 |
| **AIC** | − 1.37 | − 1.36 | − 1.38 |
| **Diagnostic tests** | | | |
| **Joint significance** | 330.95 | 210.95 | 193.26 |
| **F-Test** | (0.00) | (0.00) | (0.00) |
| **Serial correlation** | 1.94 | 2.00 | 1.96 |
| **Durbin-Watson Stat** | | | |
| **Serial correlation Breusch-Godfrey LM test** | 0.26 | 0.84 | 0.55 |
| | (0.77) | (0.43) | (0.58) |
| **Heteroskedasticity Breusch-Pagan-Godfrey test** | 1.41 | 0.90 | 2.64 |
| | (0.23) | (0.52) | (0.01) |
| **Normality test** | 25.59 | 18.33 | 90.70 |
| **Jarque-Bera stat** | (0.00) | (0.00) | (0.00) |
| **Stability diagnostic** | 0.55 | 0.46 | 0.50 |
| **Ramsey RESET test** | (0.58) | (0.63) | (0.61) |

**Note:** *p*-values are in parenthesis. ***, **, * implies statistical significance at 1 percent, 5 percent, and 10 percent, respectively. B-G LM is with two lags and Ramsey RESET test is fitted with two terms.

**Table 7. ARDL(p, q) model of SWAP5Y.**

| Dependent variable | SWAP5Y | SWAP5Y | SWAP5Y |
|---|---|---|---|
| **Main equation** | | | |
| CTB3M | 0.40*** | 0.39*** | 0.39*** |
| | (0.00) | (0.00) | (0.00) |
| CTB3M(-1) | −0.48*** | −0.46*** | −0.48*** |
| | (0.00) | (0.00) | (0.00) |
| CTB3M(-2) | 0.26** | 0.24** | 0.23** |
| | (0.01) | (0.01) | (0.01) |
| SWAP5Y(-1) | 1.11*** | 1.09*** | 1.10*** |
| | (0.00) | (0.00) | (0.00) |
| SWAP5Y(-2) | −0.25* | −0.27* | −0.25 |
| | (0.10) | (0.08) | (0.14) |
| Core CPIYOY | | 0.04 | 0.05 |
| | | (0.18) | (0.10) |
| IPYOY | | 0.004*** | 0.004*** |
| | | (0.00) | (0.00) |
| ΔLNSNGHAI | | | 0.39* |
| | | | (0.06) |
| ΔLNUSDCNY | | | −0.66 |
| | | | (0.53) |
| Intercept | 0.21*** | 0.25*** | 0.24*** |
| | (0.00) | (0.00) | (0.00) |
| **Cointegrating relationship** | | | |
| Long-term coefficient | 0.69*** | 0.48** | 0.38* |
| | (0.00) | (0.02) | (0.09) |
| Rate of adjustment | −0.15*** | −0.17*** | −0.15*** |
| | (0.00) | (0.00) | (0.00) |
| **Model information** | | | |
| Obs | 102 | 102 | 102 |
| Adj R$^2$ | 0.94 | 0.94 | 0.94 |
| AIC | − 1.37 | − 1.37 | − 1.39 |
| **Diagnostic tests** | | | |
| Joint significance | 251.85 | 192.76 | 159.50 |
| F-test | (0.00) | (0.00) | (0.00) |
| Serial correlation | 1.98 | 2.03 | 2.00 |
| Durbin-Watson stat | | | |
| Serial correlation Breusch-Godfrey LM test | 0.01 | 0.59 | 0.19 |
| | (0.99) | (0.55) | (0.83) |
| Heteroskedasticity Breusch-Pagan-Godfrey test | 1.44 | 1.36 | 2.22 |
| | (0.21) | (0.22) | (0.02) |
| Normality test | 1.68 | 1.57 | 11.56 |
| Jarque-Bera stat | (0.43) | (0.46) | (0.00) |
| Stability diagnostic | 0.25 | 0.18 | 0.11 |
| Ramsey RESET test | (0.78) | (0.84) | (0.90) |

**Note:** *p*-values are in parenthesis. ***, **, * implies statistical significance at 1 percent, 5 percent, and 10 percent, respectively. B-G LM is with two lags and Ramsey RESET test is fitted with two terms.

**Table 8. ARDL(p, q) model of SWAP10Y.**

| Dependent variable | SWAP10Y | SWAP10Y | SWAP10Y |
|---|---|---|---|
| **Main equation** | | | |
| CTB3M | 0.37*** | 0.35*** | 0.36*** |
| | (0.00) | (0.00) | (0.00) |
| CTB3M(-1) | −0.50*** | −0.48*** | −0.50*** |
| | (0.00) | (0.00) | (0.00) |
| CTB3M(-2) | 0.28** | 0.27** | 0.25** |
| | (0.01) | (0.01) | (0.01) |
| SWAP10Y(-1) | 1.24*** | 1.22*** | 1.22*** |
| | (0.00) | (0.00) | (0.00) |
| SWAP10Y(-2) | 0.38** | 0.40** | 0.39** |
| | (0.01) | (0.01) | (0.01) |
| Core CPIYOY | | 0.06* | 0.07** |
| | | (0.08) | (0.04) |
| IPYOY | | 0.004*** | 0.004** |
| | | (0.00) | (0.01) |
| ΔLNSNGHAI | | | 0.47** |
| | | | (0.02) |
| ΔLNUSDCNY | | | −0.34 |
| | | | (0.74) |
| Intercept | 0.29*** | 0.37*** | 0.35*** |
| | (0.00) | (0.00) | (0.00) |
| **Cointegrating relationship** | | | |
| Long-term coefficient | 0.55*** | 0.29 | 0.19 |
| | (0.00) | (0.13) | (0.41) |
| Rate of adjustment | −0.14*** | −0.18*** | −0.16*** |
| | (0.00) | (0.00) | (0.00) |
| **Model information** | | | |
| Obs | 102 | 102 | 102 |
| Adj R$^2$ | 0.93 | 0.93 | 0.93 |
| AIC | − 1.31 | − 1.34 | − 1.36 |
| **Diagnostic tests** | | | |
| Joint significance | 219.91 | 172.51 | 144.57 |
| F-test | (0.00) | (0.00) | (0.00) |
| Serial correlation | 1.96 | 2.02 | 2.01 |
| Durbin-Watson stat | | | |
| Serial correlation Breusch-Godfrey LM test | 0.09 | 0.42 | 0.25 |
| | (0.92) | (0.66) | (0.78) |
| Heteroskedasticity Breusch-Pagan-Godfrey test | 1.63 | 1.53 | 1.88 |
| | (0.15) | (0.66) | (0.06) |
| Normality test | 4.52 | 2.29 | 18.11 |
| Jarque-Bera stat | (0.10) | (0.31) | (0.00) |
| Stability diagnostic | 0.31 | 0.30 | 0.17 |
| Ramsey RESET test | (0.73) | (0.74) | (0.84) |

**Note:** *p*-values are in parenthesis. ***, **, * implies statistical significance at 1 percent, 5 percent, and 10 percent, respectively. B-G LM is with two lags and Ramsey RESET test is fitted with two terms.

for the sensitivity analysis are as follows:

$$CSWAPiY_t = \alpha_0 + \beta_1 CTB6M_t + \beta_2 CTB6M_{t-1} + \beta_3 CTB6M_{t-2} + \beta_4 CSWAPiY_{t-1}$$
$$+ \beta_5 CSWAPiY_{t-2} + \beta_6 CPI_t + \beta_7 IPYOY_t + \beta_8 LNSNZN300_t + \beta_9 LNEURCNY_t$$
$$+ \epsilon_t$$

where i = swap for 2-, 5-, and 10-year tenors.

The models below use the 6-month Treasury bill rate (CTB6M) instead of the CTB3M for the short-term interest rate. These models use headline inflation (CPIYOY) instead of core inflation (CCPIYOY), the percentage change in SZSE index (LNSNZN300) instead of the percentage change in SSE index (LNSNGHAI), and the percentage change in the exchange rate of the CNY against the euro (LNEURCNY) instead of the percentage change in the exchange rate of the CNY against the USD (LNUSDCNY). The only common independent variable in the two extended models is the growth of industrial production (IPYOY).

The findings in the sensitivity analysis are virtually identical to the main results. However, a vital difference is that the effect of the 6-month Treasury bill rate on the CNY swap yields is somewhat larger than the effect of the 3-month Treasury bill rate, as shown in Tables 6–8.

**Granger causality.**   Lastly, Granger causality tests are conducted to examine any presence of a reverse causality in the data. The Granger causality test results are reported in the S1 Appendix. The results shows that CNY swap yields at first difference Granger cause short-term interest rates at their first difference. However, two important caveats of the Granger causality tests need to be emphasized here. First, Granger causality tests can only be conducted on stationary time-series data. Thus, the first-differenced data have been used for these tests. In contradistinction, the ARDL models presented earlier are suitable and germane for both stationary and nonstationary time series [43]. Thus, the level variables are used for the ARDL models estimated in Tables 9–11. The ARDL models incorporate the lagged value of the dependent and independent regressors. Second, these Granger causality tests only provide some insights regarding temporal precedence of the given CNY swap yield and the short-term interest rate at their first differences; they do not provide any insight about the causal relationship between the levels of the CNY swap yield and short-term interest rate. The ARDL models estimated here are relevant for understanding the underlying dynamics of the CNY swap yield of different maturity tenors.

## Implications of the findings

These findings imply that the People's Bank of China (PBOC), China's central bank, can influence interest rate swap yields of different maturity tenors through the effects of its policy rate and monetary policy actions on the short-term interest rate. This suggest that the PBOC can sway borrowing and lending rates on a range of fixed-income instruments, including swaps and swaptions. This gives the PBOC enormous clout over China's financial system. It vindicates and extends Keynes's view that the central bank's actions have a decisive effect on the long-term interest rate in two consequential ways. First, it underpins that Keynes's hypothesis about the effect of a central bank's actions on the long-term interest rate is also applicable for over-the-counter financial instruments, such as interest rate swap yields, not just government bond yields. Second, it supports the view that Keynes's conjecture about the strong connection between the current short-term interest rate and the long-term interest rate is not merely confined to financial markets in advanced capitalist economies, such as the United States and the United Kingdom, but also holds in emerging market economies, such as China.

**Table 9. ARDL(p, q) model of SWAP2Y with alternative independent variables.**

| - | SWAP2Y | SWAP2Y | SWAP2Y |
|---|---|---|---|
| | **Main equation** | | |
| CTB6M | 0.51*** | 0.52*** | 0.51*** |
| | (0.00) | (0.00) | (0.00) |
| CTB6M(-1) | −0.52*** | −0.51*** | −0.56*** |
| | (0.00) | (0.00) | (0.00) |
| CTB6M(-2) | 0.16** | 0.14** | 0.17** |
| | (0.02) | (0.03) | (0.04) |
| SWAP2Y(-1) | 1.04*** | 1.03*** | 1.08*** |
| | (0.00) | (0.00) | (0.00) |
| SWAP2Y(-2) | −0.24* | −0.22 | −0.24 |
| | (0.09) | (0.12) | (0.17) |
| CPIYOY | | 0.01 | 0.002 |
| | | (0.54) | (0.85) |
| IPYOY | | 0.003* | 0.002 |
| | | (0.09) | (0.23) |
| ΔLNSNZN300 | | | 0.52* |
| | | | (0.09) |
| ΔLNEURCNY | | | 0.47 |
| | | | (0.59) |
| Intercept | 0.15** | 0.13* | 0.14** |
| | (0.02) | (0.05) | (0.01) |
| | **Cointegrating relationship** | | |
| Long-term coefficient | 0.81*** | 0.79*** | 0.71*** |
| | (0.00) | (0.00) | (0.00) |
| Rate of adjustment | −0.20*** | −0.19*** | −0.16*** |
| | (0.00) | (0.00) | (0.00) |
| | **Model information** | | |
| Obs | 102 | 102 | 102 |
| Adj $R^2$ | 0.94 | 0.94 | 0.94 |
| AIC | − 1.38 | − 1.36 | − 1.39 |
| | **Diagnostic tests** | | |
| Joint significance | 335.09 | 237.83 | 194.57 |
| F-test | (0.00) | (0.00) | (0.00) |
| Serial correlation | 1.97 | 1.98 | 2.01 |
| Durbin-Watson stat | | | |
| Serial correlation Breusch-Godfrey LM test | 0.23 | 0.47 | 1.46 |
| | (0.79) | (0.63) | (0.24) |
| Heteroskedasticity Breusch-Pagan-Godfrey test | 1.29 | 0.92 | 2.63 |
| | (0.28) | (0.49) | (0.01) |
| Normality test | 30.69 | 31.07 | 176.65 |
| Jarque-Bera stat | (0.00) | (0.00) | (0.00) |
| Stability diagnostic | 0.12 | 0.08 | 0.44 |
| Ramsey RESET test | (0.89) | (0.93) | (0.64) |

**Note:** *p*-values are in parenthesis. ***, **, * implies statistical significance at 1 percent, 5 percent, and 10 percent, respectively. B-G LM is with two lags and Ramsey RESET test is fitted with two terms.

**Table 10. ARDL(p, q) model of SWAP5Y with alternative independent variables.**

| Dependent variable | SWAP5Y | SWAP5Y | SWAP5Y |
|---|---|---|---|
| **Main equation** | | | |
| **CTB6M** | 0.47*** | 0.49*** | 0.45*** |
| | (0.00) | (0.00) | (0.00) |
| **CTB6M(-1)** | −0.50*** | −0.43*** | −0.52*** |
| | (0.00) | (0.00) | (0.00) |
| **CTB6M(-2)** | −0.22* | | −0.14* |
| | (0.06) | | (0.08) |
| **SWAP5Y(-1)** | 1.05*** | 0.89*** | 1.11*** |
| | (0.00) | (0.00) | (0.00) |
| **SWAP5Y(-2)** | −0.21 | | −0.22 |
| | (0.16) | | (0.22) |
| **CPIYOY** | | 0.01 | 0.003 |
| | | (0.30) | (0.79) |
| **IPYOY** | | 0.005** | 0.003 |
| | | (0.01) | (0.11) |
| **ΔLNSNZN300** | | | 0.61** |
| | | | (0.01) |
| **ΔLNEURCNY** | | | 0.53 |
| | | | (0.46) |
| **Intercept** | 0.21*** | 0.14 | 0.17** |
| | (0.00) | (0.13) | (0.01) |
| **Cointegrating relationship** | | | |
| **Long-term coefficient** | 0.73*** | 0.54* | 0.58** |
| | (0.00) | (0.08) | (0.02) |
| **Rate of adjustment** | −0.17*** | −0.11** | −0.12** |
| | (0.00) | (0.01) | (0.01) |
| **Model information** | | | |
| **Obs** | 102 | 103 | 102 |
| **Adj R$^2$** | 0.94 | 0.93 | 0.94 |
| **AIC** | − 1.39 | − 1.33 | − 1.43 |
| **Diagnostic tests** | | | |
| **Joint significance** | 256.17 | 289.00 | 183.88 |
| **F-test** | (0.00) | (0.00) | (0.00) |
| **Serial correlation** | 1.96 | 1.69 | 2.06 |
| **Durbin-Watson stat** | | | |
| **Serial correlation Breusch-Godfrey LM test** | 0.05 | 1.38 | 0.73 |
| | (0.96) | (0.26) | (0.49) |
| **Heteroskedasticity Breusch-Pagan-Godfrey test** | 1.04 | 0.78 | 2.09 |
| | (0.40) | (0.56) | (0.04) |
| **Normality test** | 4.09 | 4.03 | 38.10 |
| **Jarque-Bera stat** | (0.13) | (0.13) | (0.00) |
| **Stability diagnostic** | 0.01 | 0.15 | 0.18 |
| **Ramsey RESET test** | (0.99) | (0.86) | (0.84) |

**Note:** *p*-values are in parenthesis. ***, **, * implies statistical significance at 1 percent, 5 percent, and 10 percent, respectively. B-G LM is with two lags and Ramsey RESET test is fitted with two terms.

**Table 11. ARDL(p, q) model of SWAP10Y with alternative independent variables.**

| Dependent variable | SWAP10Y | SWAP10Y | SWAP10Y |
|---|---|---|---|
| **Main equation** | | | |
| CTB6M | 0.42*** | 0.42*** | 0.42*** |
| | (0.00) | (0.00) | (0.00) |
| CTB6M(-1) | −0.50*** | −0.49*** | −0.56*** |
| | (0.00) | (0.00) | (0.00) |
| CTB6M(-2) | 0.17** | 0.15** | 0.20*** |
| | (0.01) | (0.02) | (0.00) |
| SWAP10Y(-1) | 1.21*** | 1.19*** | 1.26*** |
| | (0.00) | (0.00) | (0.00) |
| SWAP10Y(-2) | −0.36** | −0.34** | −0.38** |
| | (0.02) | (0.02) | (0.02) |
| CPIYOY | | 0.01 | 0.001 |
| | | (0.55) | (0.95) |
| IPYOY | | 0.003** | 0.002 |
| | | (0.02) | (0.14) |
| ΔLNSNZN300 | | | 0.67*** |
| | | | (0.00) |
| ΔLNEURCNY | | | 1.09* |
| | | | (0.08) |
| Intercept | 0.28*** | 0.25** | 0.26*** |
| | (0.05) | (0.01) | (0.00) |
| **Cointegrating relationship** | | | |
| Long-term coefficient | 0.59*** | 0.55*** | 0.45* |
| | (0.00) | (0.00) | (0.06) |
| Rate of adjustment | −0.15*** | −0.14*** | −0.12*** |
| | (0.00) | (0.00) | (0.00) |
| **Model information** | | | |
| Obs | 102 | 102 | 102 |
| Adj $R^2$ | 0.93 | 0.93 | 0.93 |
| AIC | −1.33 | −1.31 | −1.39 |
| **Diagnostic tests** | | | |
| Joint significance | 266.12 | 189.18 | 163.21 |
| F-test | (0.00) | (0.00) | (0.00) |
| Serial correlation | 1.99 | 2.00 | 2.07 |
| Durbin-Watson stat | | | |
| Serial correlation Breusch-Godfrey LM test | 0.04 | 0.15 | 0.79 |
| | (0.96) | (0.86) | (0.45) |
| Heteroskedasticity Breusch-Pagan-Godfrey test | 2.39 | 1.63 | 1.78 |
| | (0.04) | (0.14) | (0.08) |
| Normality test | 7.00 | 8.04 | 57.01 |
| Jarque-Bera stat | (0.03) | (0.02) | (0.00) |
| Stability diagnostic | 0.35 | 0.33 | 0.01 |
| Ramsey RESET test | (0.71) | (0.72) | (0.99) |

**Note:** *p*-values are in parenthesis. ***, **, * implies statistical significance at 1 percent, 5 percent, and 10 percent, respectively. B-G LM is with two lags and Ramsey RESET test is fitted with two terms.

## Conclusion

The empirical findings presented in this paper illustrate that the short-term interest rate has an economically and statistically significant effect on the CNY-denominated long-term interest rate swap yield, after controlling for key macroeconomic variables, such as inflation or core inflation, the growth of industrial production, the percentage change in the equity market index, and the percentage change in the exchange rate of the currency. Three different models of the long-term swap yield of different maturity tenors are estimated to show these results are quite robust, irrespective of the specifications of the estimated regression equation. Alternative choices of independent variables bare that the empirical findings are well grounded and not sensitive to or dependent on the choice of variable.

There is a lacuna in the empirical modeling of interest rate swap yields in emerging markets. The financial sector already plays a stalwart role in the Chinese economy. However, with the rise of the financial sector and financialization in emerging markets such as China, interest rate swaps are likely to play an increasingly important role in the financial systems of these economies. This paper fills a critical gap in the empirical modeling of interest rate swap yields in China. The empirical modeling of interest rate swaps in China and other emerging markets can enable policymakers and investors to go beyond just understanding the dynamics of swaps yields to illuminate the workings of the financial system and capital markets, and assess the effectiveness of the monetary transmission mechanism.

## Supporting information

**S1 Dataset.**
(XLSX)

**S1 Appendix.**
(DOCX)

## Acknowledgments

The authors thank Dr. Ricky Chee Jiun Chia (Academic Editor, *PLOS ONE*), Dr. Rabia Arif (referee), and an anonymous referee for their valuable comments on an earlier version of the paper. The authors also thank Ms. Elizabeth Dunn for her copyediting support.

**Disclaimer:** The authors' institutional affiliations are provided solely for identification purposes. Views expressed are solely those of the authors. The standard disclaimer holds.

## Author Contributions

**Conceptualization:** Tanweer Akram, Khawaja Mamun.

**Data curation:** Tanweer Akram.

**Formal analysis:** Tanweer Akram, Khawaja Mamun.

**Investigation:** Tanweer Akram, Khawaja Mamun.

**Methodology:** Tanweer Akram, Khawaja Mamun.

**Project administration:** Tanweer Akram, Khawaja Mamun.

**Supervision:** Tanweer Akram.

**Validation:** Tanweer Akram.

**Visualization:** Khawaja Mamun.

**Writing – original draft:** Tanweer Akram, Khawaja Mamun.

**Writing – review & editing:** Tanweer Akram, Khawaja Mamun.

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
