## [Decision Letter · Decision Letter 0]

18 May 2023

PONE-D-23-10931Chinese Yuan Interest Rate Swap YieldsPLOS ONE

Dear Dr. Tanweer Akram,

Thank you for submitting your manuscript to PLOS ONE. After careful consideration, we feel that it has merit but does not fully meet PLOS ONE’s publication criteria as it currently stands. Therefore, we invite you to submit a revised version of the manuscript that addresses the points raised during the review process.

We look forward to receiving your revised manuscript.

Kind regards,

Ricky Chee Jiun Chia

Academic Editor

PLOS ONE

Journal Requirements:

Reviewers' comments:

Reviewer's Responses to Questions

**Comments to the Author**

1. Is the manuscript technically sound, and do the data support the conclusions?

Reviewer #1: Partly

Reviewer #2: Yes

2. Has the statistical analysis been performed appropriately and rigorously? 

Reviewer #1: Yes

Reviewer #2: No

3. Have the authors made all data underlying the findings in their manuscript fully available?

Reviewer #1: Yes

Reviewer #2: No

4. Is the manuscript presented in an intelligible fashion and written in standard English?

Reviewer #1: Yes

Reviewer #2: Yes

5. Review Comments to the Author

Reviewer #1: The study examines the relationship between Chinese yuan – denominated long-term interest rate swap yields. The authors can capture the relationship through the application of autoregressive distributed lag (ARDL) model. Their results shows that the yield of the three-month Treasury bills has a positive and statistically significant effect on the swap yield and the long-run relationship varies significantly from the two-year maturity term to ten-year maturity. This suggest that the People’s Bank of China (PBOC) can influence borrowing and lending rates on a range of fixed-income instruments, including swaps and swaptions.

I only have few comments below:

1. The authors stated that the null hypotheses (results presented in 2a and 2b) for the ADF and the KPSS tests are different, they should briefly explain the differences in the null hypotheses.

2. First paragraph of the Econometric Results section: authors should describe the meaning of "basis point", even in a footnote. This will help those that are not familiar with the financial language.

3. The authors should add a brief description of their econometric model.

4. Including one or two charts and a summary statistic table will help to present their data to the readers.

Reviewer #2: The article "Chinese Yuan Interest Rate Swap Yields” focuses on analyzing the dynamics of Chinese Yuan Interest Rate Swap (CNY IRS) yields. The authors examine the volatility and relationships between CNY IRS yields and other macroeconomic variables such as inflation, exchange rate, and money supply. They use daily data from 2014 to 2022 and employ Auto regressive distributed lag (ARDL) technique to estimate the volatility and co-movements between variables more specifically to estimate the impact of short-term swap yields on the long-term interest rate swap yields. Long-term interest swap yields are a valuable tool for monitoring economic trends and informing policy and investment decisions especially for a country like China, given the country's rapidly growing economy and increasing integration into global financial markets. By examining these yields, policymakers and investors can gain valuable insights into the future direction of interest rates and broader economic trends, which can help them to make more informed decisions and manage risk more effectively.

However, I believe that this article should only be accepted after incorporating major changes that are as follows:

1. To make introduction coherent the authors should incorporate following information:

• General picture of China’s economy and link it to the relevance of interest rate swap yields using statistics (include graphs).

• Background of the interest swap yields in China and how have the authors calculated the yields in the literature versus what authors have used in their analysis (if it is any different or same).

• Main results of the study.

2. Expand on literature by including different methodologies that researchers have used to estimate the following relationship and compare it to the one used in papers. Last paragraph should discuss the major contributions of the study. Also cite some studies that have used interest rate swap yields in conducting different analysis for different countries generally and specifically for China.

3. Include theoretical framework.

4. Include empirical equation and explain the methodology in detail.

5. Explain why the specific time period is selected. Also, during this time period a lot of changes have happened in China kindly explain the relevance of those structural changes that may have an impact on the estimated coefficients.

6. Sensitivity analysis is missing from the paper.

7. To check for the reverse causality that may exist in the relation, the authors should report the granger causality test and explain how they have dealt with the problem of reverse causality in the estimations.

8. The tables should be edited properly so that they become reader friendly and it becomes easy to read the dependent variable and independent variables easily. Put down all the other important variables in the notes underneath the tables that were controlled in the regressions.

6. PLOS authors have the option to publish the peer review history of their article (what does this mean?). If published, this will include your full peer review and any attached files.

Reviewer #1: No

Reviewer #2: **Yes: **rabia arif

---

## [Author Response · Author response to Decision Letter 0]

14 Jul 2023

PONE-D-23-10931

Chinese Yuan Interest Rate Swap Yields

Dr. Ricky Chee Jiun Chia

Academic Editor

PLOS ONE 

July 13, 2023

Dear Dr Chia:

We thank the editor and the two referees for their valuable comments. These comments have been extremely useful for us in improving the paper. Hence, we have revised the manuscript, incorporating the referees’ suggestions and addressing the issues they raised. We hope that you will find the revised paper suitable for publication in PLOS ONE. The revised parts of the paper are highlighted in yellow.

A separate note responds to each point raised by Referee 1 and Referee 2, stating how we have revised the manuscript in accordance with their comments.

Sincerely,

The authors

*******

 

RESPONSE TO REFEREE 1 (ANONYMOUS)

We thank Referee 1 for the detailed comments. We have revised the paper based on both referees’ valuable suggestions. The revised parts of the paper are highlighted in yellow in the manuscript. Our responses to Referee 1’s comments are provided below.

Referee 1’s comment 1: “The authors stated that the null hypotheses (results presented in [Tables] 2a and 2b) for the ADF and the KPSS tests are different, they should briefly explain the differences in the null hypotheses.”

Authors’ response: We have explicitly stated the different null hypotheses for the ADF and the KPSS tests in the revised paper (p. 16).

Referee 1’s comment 2: “First paragraph of the Econometric Results section: authors should describe the meaning of ‘basis point,’ even in a footnote. This will help those that are not familiar with the financial language.”

Authors’ response: We have explained the meaning of ‘basis point’ in the text in the revised paper (p. 19).

Referee 1’s comment 3: “The authors should add a brief description of their econometric model.”

Authors’ response: We have added a brief description of the econometric model in the revised paper (pp. 10-12).

Referee 1’s comment 4: “Including one or two charts and a summary statistic table will help to present their data to the readers.”

Authors’ response: We have added three charts. We has also added summary tables (Tables 2A and 2B) that provide summary statistics in the revised paper (pp. 3, 5, 15-16).

Authors’ additional remarks: We have extended the dataset in the revised paper. We also addressed Referee 2’s comments. 

 

RESPONSE TO REFEREE 2 (DR. RABIA ARIF)

We thank Dr. Rabia Arif (Referee 2) for her detailed comments. We have revised the paper based on both referees’ valuable suggestions. The revised parts of the paper are highlighted in yellow in the manuscript. Our responses to Referee 2’s detailed comments are provided below.

Referee 2’s general comment on Data availability: “Have the authors made all data underlying the findings in their manuscript fully available? No.”

Authors’ response: We have included the dataset with the paper in an Excel spreadsheet, S1_Dataset.xlxs. In the revised version of the paper, we have also included summary statistics and other additional information. 

Referee 2’s comment 1: “To make introduction coherent the authors should incorporate following information: (1) General picture of China’s economy and link it to the relevance of interest rate swap yields using statistics (include graphs). (2) Background of the interest swap yields in China and how have the authors calculated the yields in the literature versus what authors have used in their analysis (if it is any different or same). (3) Main results of the study.”

Authors’ response: We have revised the introduction in the updated version of the paper. The revised introduction provides a brief overview of the Chinese economy and recent developments therein. We also added some background information on the use of CNY swaps and summarized the main results of the study in the introduction, as suggested by the referee (pp. 2-5).

Referee 2’s comment 2: “Expand on literature by including different methodologies that researchers have used to estimate the following relationship and compare it to the one used in papers. Last paragraph should discuss the major contributions of the study. Also cite some studies that have used interest rate swap yields in conducting different analysis for different countries generally and specifically for China.”

Authors’ response: We discuss the different approaches that researchers have used in their attempts to model the dynamics of swap yields. We argue that the Keynesian perspective is a fruitful approach to modeling the dynamics of swap yields. We cite several studies that have modeled interest rate swap yields in different countries. An extensive research in the major databases, including JSTORE, ECONLIT, Google Scholar, PROQUEST, SSRN, and so forth, shows that there are no econometric studies of CNY-denominated interest rate swap yields. This paper fills a critical gap in the literature (pp. 6-10)

Referee 2’s comment 3: “Include theoretical framework.”

Authors’ response: We have provided a theoretical framework and illustrated the theoretical framework with a simple quantitative model of the dynamics of swap yields based on key macroeconomic variables. All the additions are presented in yellow highlight (pp. 11-12).

Referee 2’s comment 4: “Include empirical equation and explain the methodology in detail.”

Authors’ response: In the revised paper we have included the empirical equations and explained the methodology in detail (pp. 18, 24).

Referee 2’s comment 5: “Explain why the specific time period is selected. Also, during this time period a lot of changes have happened in China. Kindly explain the relevance of those structural changes that may have an impact on the estimated coefficients.”

Authors’ response: The time period is based on: (1) the introduction and widespread use of CNY-denominated interest rate swaps for both hedging and speculating on interest rate risks by domestic and overseas parties, and (2) the availability of the data. Please note that the dataset has been extended in the revised version to include the latest data available as of June 2023.

Referee 2’s comment 6: “Sensitivity analysis is missing from the paper.”

Authors’ response: We have provided robust checks to our estimates by using a different set of independent variables. Specifically we have used 6-month Treasury bills’ rates instead of 3-month Treasury bills’ rates for the short-term interest rate; total inflation instead of core inflation; Shenzhen equity price index instead of Shanghai equity price index; and the exchange rate of the Chinese yuan against the euro instead of the US dollar. The econometric results are quite similar across the swap yield curve. Moreover, it should be pointed out that for each of the dependent variables, namely the 2-year, 5-year, and 10-year swap yields, three different models are estimated. All of these models show that the short-term interest rate has a statistically significant and economically important effect on the swap yield. This implies that the econometric findings are robust to changes in the functional forms of the regression models. The results of the sensitivity analysis are presented in Tables 6A, 6B, and 6C (pp. 24-27).

Referee 2’s comment 7: “To check for the reverse causality that may exist in the relation, the authors should report the granger causality test and explain how they have dealt with the problem of reverse causality in the estimations.”

Authors’ response: We have undertaken Granger causality tests. The test results are presented in Appendix A (p. 29). We also discuss how the ARDL models address reverse causality in our estimations (p. 28).

Referee 2’s comment 8: “The tables should be edited properly so that they become reader friendly and it becomes easy to read the dependent variable and independent variables easily.” 

Authors’ response: We have revised the tables by breaking them into three separate tables (see Tables 5A, 5B, and 5C) for different swap maturity tenors to present the econometric results (pp. 20-22).

Authors’ additional remarks: We have extended the dataset. We also addressed Referee 1’s comments.

---

## [Editor Report · Decision Letter 1]

25 Jul 2023

Chinese Yuan Interest Rate Swap Yields

PONE-D-23-10931R1

Dear Dr. Tanweer Akram,

We’re pleased to inform you that your manuscript has been judged scientifically suitable for publication and will be formally accepted for publication once it meets all outstanding technical requirements.

Kind regards,

Ricky Chee Jiun Chia

Academic Editor

PLOS ONE
---

## [Editor Report · Acceptance letter]

27 Jul 2023

PONE-D-23-10931R1 

Chinese Yuan Interest Rate Swap Yields 

Dear Dr. Akram:

I'm pleased to inform you that your manuscript has been deemed suitable for publication in PLOS ONE. Congratulations! Your manuscript is now with our production department. 

Kind regards, 

on behalf of

Dr. Ricky Chee Jiun Chia 

Academic Editor

PLOS ONE